# DISENTANGLED CUMULANTS HELP SUCCESSOR REPRESENTATIONS TRANSFER TO NEW TASKS

## ABSTRACT

Biological intelligence can learn to solve many diverse tasks in a data efficient manner by re-using basic knowledge and skills from one task to another. Furthermore, many of such skills are acquired without explicit supervision in an intrinsically driven fashion. This is in contrast to the state-of-the-art reinforcement learning agents, which typically start learning each new task from scratch and struggle with knowledge transfer. In this paper we propose a principled way to learn a basis set of policies, which, when recombined through generalised policy improvement, come with guarantees on the coverage of the final task space. In particular, we concentrate on solving goal-based downstream tasks where the execution order of actions is not important. We demonstrate both theoretically and empirically that learning a small number of policies that reach intrinsically specified goal regions in a disentangled latent space can be re-used to quickly achieve a high level of performance on an exponentially larger number of externally specified, often significantly more complex downstream tasks. Our learning pipeline consists of two stages. First, the agent learns to perform intrinsically generated, goal-based tasks in the total absence of environmental rewards. Second, the agent leverages this experience to quickly achieve a high level of performance on numerous diverse externally specified tasks.

## 1 INTRODUCTION

Natural intelligence is able to solve many diverse tasks by transferring knowledge and skills from one task to another. For example, by knowing about objects and how to move them in 3D space, it is possible to learn how to sort them by shape or colour faster. However, many of the current state-of-the-art artificial reinforcement learning (RL) agents often struggle with such basic skill transfer. They are able to solve single tasks well, often beyond the ability of any natural intelligence (Silver et al., 2016; Mnih et al., 2015; Jaderberg et al., 2017), however even small deviations from the task that the agent was trained on can result in catastrophic failures (Lake et al., 2016; Rusu et al., 2016). Although improving transfer in RL agents is an active area of research (Higgins et al., 2017a; Rusu et al., 2016; Nair et al., 2018; Barreto et al., 2018; Wulfmeier et al., 2019; Torrey & Shavlik, 2010; Taylor & Stone, 2009; Thrun & Pratt, 2012; Caruana, 1997; Jaderberg et al., 2017; Riedmiller et al., 2018), most typical deep RL agents start learning every task from scratch. This means that each time they have to re-learn how to perceive the world (the mapping from a high-dimensional observation to state), and also how to act (the mapping from state to action), with the majority of time arguably spent on the former. The optimisation procedure naturally discards information that is irrelevant to the task, which means that the learnt state representation is often unsuitable for new tasks. Biological intelligence appears to operate differently. A lot of knowledge tends to be discovered and learnt without explicit supervision (Tolman, 1948; Clark, 2013; Friston, 2010). This basic knowledge can then form the behavioural basis that can be used to solve new tasks faster. In this paper we argue that such transferable knowledge and skills should be acquired in artificial agents too. In particular, we want to start by building agents that have the ability to discover stable entities that make up the world and to learn basic skills to manipulate these entities. Compositional re-use of such skills enables biological intelligence to find reasonable solutions to many naturally occurring tasks, from goal-directed movement (controlling your own position), to food gathering (controlling the position of fruit and and nuts), or building a simple defence system (re-positioning multiple stones into a fence or digging a ditch). In this paper we concentrate on goal-based natural tasks that can be expressed in natural language, and that do not require a specific execution order of actions.

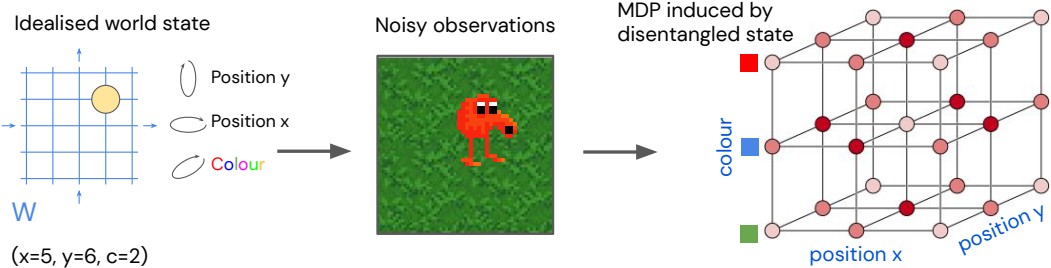

Figure 1: The idealised world state completely described by compositions of the following independent transformations: changes in position x, y and colour. Such a state may be projected into a high-dimensional observation, which may contain a lot of irrelevant detail, like the particular instantiation of the Qbert, or the grassy background. Disentangled representations recover the meaningful information about the independently transformable aspects of the world and disregard the irrelevant details.

To this end, we propose a principled way to learn a small set of policies which can be re-used by the agents to quickly produce reasonable performance on an exponentially large set of goal-driven tasks within an environment. We propose a method on how to discover these policies in the absence of external supervision, where the agent accumulates a transferable set of basic skills through intrinsically motivated interactions with the environment. This first stage of free play builds the foundation to later solve many diverse extrinsically specified downstream tasks. We suggest formalising such a two-stage pipeline as the *endogenous reinforcement learning* (ERL) setting, in order to provide a consistent evaluation framework for some of the existing and future work on building RL agents with intrinsic learning signals (Gregor et al., 2017; Eysenbach et al., 2019; Hansen et al., 2019; Nair et al., 2018; Laversanne-Finot et al., 2018). We propose a disjoint two step research pipeline, where the agent is allowed unlimited access to the environment in the ERL stage, where no extrinsic rewards are provided and the agent is supposed to learn as much as it can through endogenously (intrinsically) driven interactions with the environment. This is followed by a standard RL stage where the success of the previous step is evaluation in terms of data-efficiency of learning on multiple diverse extrinsically (exogenously) specified downstream tasks in the same environment. We hope that by working in this extreme two stage setting, where the agents have to learn useful knowledge with no access to task rewards, we can develop algorithms that learn more robust and transferable policies even in the traditional RL setting.

In this paper we propose to use the ERL stage to discover disentangled features through task-free interactions with the environment, and then solving a number of goal-driven self-generated tasks specified in the learnt disentangled feature space. In particular, we suggest learning $k$ disentangled features, discretising them into $m$ bins each, and learning $km$ feature control policies that achieve the respective bin value of the given feature. We then re-combine the feature control policies learnt in the ERL stage to solve downstream tasks in the RL stage in a few-shot manner using Generalized Policy Improvement (GPI) (Barreto et al., 2018). We demonstrate theoretically and empirically that our proposed set of basis policies that learn to control disentangled features produce significantly better generalisation over a large number of downstream tasks.

Intuitively, disentangled representations consist of the smallest set of features that represent those aspects of the world state that are independently affected by natural transformations and together explain the most of the variance observed in an environment (Higgins et al., 2018) (see Fig. 1). Disentangled features, therefore, carve the world at its joints and provide a parsimonious representation of the world state that also points to which aspects of the world are stable, and which can in principle be transformed independently of each other. We conjecture that disentangled features align well with the idealised state space in which natural tasks are defined. Hence, by learning a set of policies that can control these features an agent will acquire a set of basis policies which spans a large set of natural tasks defined in such an environment. Note that both disentangled features and their respective control policies can be learnt without an externally specified task, purely in the ERL setting. We provide both theoretical justification for this setup, as well as experimental illustrations of the benefit of disentangled representations in a large set of tasks of varying difficulty.

Hence, the main contribution of this work is a theoretical result that extends the GPI framework to guarantee achievability on a large set of natural goal-driven tasks given a small set of basis policies that control disentangled features. In particular, we demonstrate that given $k$ disentangled features discretised into $m$ bins, we can guarantee achievability with a deterministic policy on at least $(m+1)^k$ downstream tasks by using GPI to recombine $km$ feature control policies discovered and learnt purely through intrinsically driven interactions with the environment in the total absence of environment rewards. Our result holds for any tasks that can be specified in natural language and do not require a particular ordering of the actions to be solved. For example, our approach would be able to solve a task that requires sorting objects in space based on their colour or shape, or tidying up a messy playroom by putting all the toys in a box, but it will not be able to solve a task like cooking a meal, where the execution order of the different stages in the recipe matters.

**Related work**   Past work on supervised and reinforcement learning has demonstrated how multi-task learning, transfer and adaptation can provide strong performance gains across various domains (Caruana, 1997; Thrun & Pratt, 2012; Yosinski et al., 2014; Girshick et al., 2014; Jaderberg et al., 2017; Riedmiller et al., 2018; Wulfmeier et al., 2019). Typically these approaches use hand-crafted auxiliary tasks to boost learning of the downstream tasks of interest, which is not scalable and comes with no guarantees on which set of auxiliary tasks is optimal for boosting performance on a large number of downstream tasks. A number of other past approaches shared our motivation of replacing the hand-crafted auxiliary task specification by an automatic way of discovering a diverse and useful set of policies in the absence of externally specified tasks. The predominant approach so far has been to optimise an objective that encourages behaviours that are both diverse and distinguishable from each other (Gregor et al., 2017; Eysenbach et al., 2019; Hansen et al., 2019), or to learn how to solve intrinsic tasks sampled from a learnt representation space (Nair et al., 2018; Laversanne-Finot et al., 2018). While these approaches have been shown to be successful on transferring the learnt policies to solve certain downstream tasks, none of them provided theoretical guarantees on the downstream task coverage by the basis set of policies. Such guarantees were however provided by van Niekerk et al. (2018) and Barreto et al. (2017; 2018). These papers calculated how well a given set of policies can be transferred to solve a wide range of downstream tasks. However, they left the question of how to discover such a set of basis policies open. Hence, our work provides a unique perspective by addressing both the questions of what makes a good basis set of policies to get certain guarantees on final task coverage, and how these policies may be learnt in the ERL setting. Other related literature worth noting is the work by Higgins et al. (2017b), who showed that learning a downstream task policy over disentangled representations improved its robustness to visual changes in the environment. Another piece of work (Machado et al., 2018) demonstrated the usefulness of discovering reward-agnostic options through successor feature learning for improving data efficiency in downstream task learning. The benefits of these options, however, were primarily through improving exploration. No guarantees were given in terms of downstream task coverage.

## 2   BACKGROUND

**Basic Reinforcement Learning (RL) formalism.**   An RL agent interacts with its environment through a sequence of actions in such a way as to maximise the expected cumulative discounted rewards (Sutton & Barto, 1998). The RL problem is typically expressed using the formalism of Markov Decision Processes (MDPs) (Puterman, 1994). An MDP is a tuple $M = (\mathcal{S}, \mathcal{A}, \mathcal{P}, \mathcal{R}, \gamma)$, where $\mathcal{S}$ and $\mathcal{A}$ are the sets of states and actions, $\mathcal{P}$ is the transition probability that predicts the distribution over next states given the current state and action $s' \sim \mathcal{P}(\cdot|s,a)$, $\mathcal{R}$ is the distribution of rewards $r \sim \mathcal{R}(s,a,s')$ received for making the transition $s \overset{a}{\mapsto} s'$, and $\gamma \in [0,1)$ is the discount factor used to make future rewards progressively less valuable. Given an MDP, the goal of the agent is to maximise the expected return $G_t = \sum_{i=0}^{\infty} \gamma^i r_{t+i}$. This is done by learning a policy $\pi(a|s)$ that selects the optimal action $a \in \mathcal{A}$ in each state $s \in \mathcal{S}$. A typical RL problem attempts to find the optimal policy $\pi^* = \underset{\pi}{\text{argmax}} \, \mathbb{E}\left[\sum_{t \geq 0} \gamma^t r | \pi\right]$, where the expectation is taken over all possible interaction sequences of the agent's policy with the environment. The optimal policy is learnt with respect to a particular task operationalised through the choice of the reward function $\mathcal{R}(s,a,s')$.

**Successor Features**   The successor feature (SF) representation is a way of decoupling the dynamics of an environment from its reward function. This is done by representing an environment reward

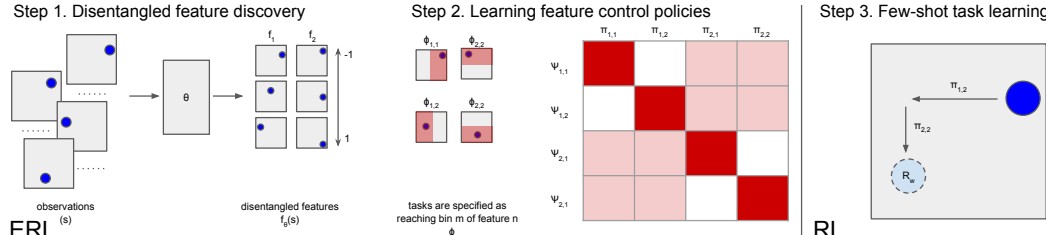

Figure 2: Schematic illustration of the three steps of our method. First we use an existing method for unsupervised disentangled feature discovery from observations obtained using an exploration policy. We then learn control policies that learn to achieve certain uniformly spread values for the learnt features. Finally, we use the feature control policies to solve tasks using the GPI framework. The first two steps do not require any extrinsic rewards.

as $r(s, a, s') = \phi(s,a,s')^\top \boldsymbol{w}$ where $\phi(s,a,s')$ is a vector of environment features. Notably, this representation does not decrease the expressivity of $r$ since no assumptions are made on the form of $\phi$. Moreover, this representation allows for decomposing value functions as follows:

$$Q^\pi(s,a) = \mathbb{E}^\pi \left[ \sum_{k=0}^\infty \gamma^k \phi(s_t,a_t,s_{t+1}) | s_0 = s, a_t = a \right]^\top \boldsymbol{w}_j = \psi(s,a)^{\pi\top} \boldsymbol{w}_j. \tag{1}$$

where $\psi(s,a)^\pi$ is a vector of reward-independent *successor features*.

**GPI & GPE**   Generalised Policy Improvement (GPI) and Generalised Policy Evaluation (GPE) (Barreto et al., 2017) can be used to transfer a set of existing policies to solve new tasks. The framework is specified for a set of MDPs:

$$\mathcal{M}^\phi(\mathcal{S},\mathcal{A},\mathcal{P},\gamma) = \{ M^\phi(\mathcal{S},\mathcal{A},\mathcal{P},r,\gamma) \mid r(s,a,s') = \phi(s,a,s')^\top \boldsymbol{w} \} \tag{2}$$

induced by all possible choices of weights $\boldsymbol{w}$ that specify all possible rewards $r$, given a state space $\mathcal{S}$, action space $\mathcal{A}$, transition probabilities $\mathcal{P}$, discount factor $\gamma$ and features $\phi(s,a,s')$. Note that the features are meant to be the same for all MDPs $M \in \mathcal{M}^\phi$. Given a policy $\pi_i$ learnt to solve task $i$ specified by $\boldsymbol{w}_i$, we can evaluate its value under a different reward $r_j = \phi(s,a,s')^\top \boldsymbol{w}_j$ using GPE:

$$Q_j^{\pi_i}(s,a) = \psi(s,a)^{\pi_i\top} \boldsymbol{w}_j \tag{3}$$

using our definition of successor features defined in (1).

Hence, given a set of policies $\pi_1$, $\pi_2$, ..., $\pi_i$ induced by rewards $r_1$, $r_2$, ..., $r_i$ over a subset of the MDPs $M' \subset \mathcal{M}^\phi$, we can get a new policy $\pi_j$ for a new task induced by $r_j$ (note that $M_j \in \mathcal{M}^\phi$, $M_j \cap M' = \emptyset$) according to:

$$\pi_j(s) = \underset{a}{\operatorname{argmax}} \max_i \psi(s,a)^{\pi_i\top} \boldsymbol{w}_j. \tag{4}$$

## 3   ENDOGENOUS RL WITH GPE AND GPI

This section provides a general overview of the proposed ERL pipeline, consisting of (1) a representation learning phase, (2) an intrinsic reinforcement learning phase and (3) a few-shot learning phase when a new task is presented, where steps 1-2 form the ERL stage, and step 3 forms the RL stage (see Fig. 2). In Sec. 4 we will present the main theoretical contribution of our paper where we will discuss why a particular version of the pipeline that uses *disentangled* features is expected to perform well in the RL stage.

**Representation learning.**   At the beginning of the pipeline our agents learn a parameterized representation of the environment's state which we use to define a set of features $\phi_{1:n}(s) \in \Phi \subseteq \mathbb{R}^n$.

Each feature $\phi_i$ will be used as a cumulant that gives rise to an RL task. Specifically, each feature $\phi_i$ will induce an associated optimal policy $\pi_i$ that maximises the expected discounted sum of $\phi_i$.

We propose defining features $\phi_i$ in the following way. Let $f_{1:k} : \mathcal{S} \mapsto [0,1]^k$ be an arbitrary continuous function. We define a discretization of each $f_i(s)$ into $m$ uniformly spaced bins: $[0,1/m),[1/m,2/m),...,[(m-1)/m,1)$, denoted $b_1,...,b_m$ respectively. Using this notation, we can define a set of $n = mk$ features as follows:

$$\phi_{ij}(s,a) = \mathbf{1}\{f_i(s) \in b_j\}, \tag{5}$$

where $\mathbf{1}\{\cdot\}$ is the indicator function. Note that, to facilitate the exposition, we use two indices to refer to a specific feature; obviously, we can "flatten" these indices and treat the features as a vector if needed. Intuitively, we can think of the task induced by $\phi_{ij}(s,a)$ as setting the $i$-th representational latent dimension to a value in the interval $b_j$. Specific choices of $m$ and $k$ in our experiments are explained in the supplementary material.

**Intrinsic RL**   As discussed above, each feature $\phi_{ij}$ gives rise to an RL task. The second stage of our pipeline consists of solving these tasks. Crucially, instead of computing the value function of policy $\pi_{ij}^*$ with respect to cumulant $\phi_{ij}$ only, we will compute the successor features of $\pi_{ij}^*$ with respect to *all* cumulants:

$$\psi_{lh}^{\pi_{ij}^*}(s,a) = \mathbb{E}_{\pi_{ij}^*}\left[\sum_{t=0}^{\infty}\gamma^t\phi_{lh}(s_t,a_t)|s_0=s,a_0=a\right] \tag{6}$$

where $\pi_{ij}^*(s)$ is one of the optimal policies induced by cumulant $\phi_{ij}$, that is, $\pi_{ij}^*(s) \in \operatorname{argmax}_\pi Q_{ij}^\pi(s,\pi(s))$.

Collectively the successor features $\psi_{lh}^{\pi_{ij}^*}(s,a)$ can be thought of as an $n \times n$ matrix $\mathbf{\Psi}$ with cumulants in one dimension and policies in the other dimension. The matrix $\mathbf{\Psi}$ represents the agent's knowledge of how to manipulate the features of the environment. We will also use our double-subscript notation to refer to specific elements of $\mathbf{\Psi}(s,a)$: $\mathbf{\Psi}_{(lh),(ij)}(s,a) = \psi_{lh}^{\pi_{ij}^*}(s,a)$.

**Few-shot learning phase**   Provided our agent has invested an initial effort to accurately learn the matrix $\mathbf{\Psi}$, we can now leverage it to perform few-shot learning. First, note that any linear combination of cumulants $\phi_{1:n}$, $\phi_w = \sum_i w_i\phi_i$, is itself a cumulant. We can then define the set of cumulants

$$\Phi = \left\{\phi_w(s,a) = \sum_{i,j} w_{ij}\phi_{ij}(s,a) \,|\, w \in \mathbb{R}^{k \times m}\right\}. \tag{7}$$

Given an arbitrary task, we can find a cumulant $\phi_w \in \Phi$ that approximates the task as well as possible. One way to do so is to select $w \in \mathbb{R}^{k \times m}$ such that

$$w = \operatorname*{argmin}_{w' \in \mathbb{R}^{k \times m}} \mathbb{E}_{(s,a)\sim\mathcal{D}}\left[||\sum_{ij} w'_{ij}\phi_{ij}(s,a) - r(s,a)||\right], \tag{8}$$

where $\mathcal{D}$ is a distribution over $\mathcal{S} \times \mathcal{A}$ and $||\cdot||$ is a norm. Note that (8) is a linear regression.

Once we have computed $w \in \mathbb{R}^{k \times m}$, we can use the successor features of policy $\pi_{ij}^*$ to evaluate it on task $\phi_w$ (a process sometimes referred to as Generalized Policy Evaluation or GPE):

$$Q_w^{\pi_{ij}^*}(s,a) = \sum_{l,h} w_{lh}\psi_{lh}^{\pi_{ij}^*}(s,a), \tag{9}$$

where $Q_w^{\pi_{ij}^*}(s,a)$ is the action-value function of $\pi_{ij}^*$ under cumulant $\phi_w$. Finally, by using GPI we can directly compute a policy based on the known policies $\pi_{ij}^*$:

$$\pi_w^{\text{GPI}}(s) = \operatorname*{argmax}_a \max_{ij} Q_w^{\pi_{ij}^*}(s,a). \tag{10}$$

While $\phi_w$ is not guaranteed to be optimal with respect to $\phi_w$, its performance on this task is at least as good, and generally better, than that of the known policies $\pi_{ij}^*$ (Barreto et al., 2017). Moreover, the computation of $\pi_w^{\text{GPI}}$ is immediate given the matrix $\mathbf{\Psi}$ and $w$. Since we assume the matrix $\mathbf{\Psi}$ has been pre-computed in the ERL phase, this essentially reduces an RL problem to the regression problem shown in (8).

## 4 THEORETICAL RESULTS

This section forms the main theoretical contribution of this paper. We highlight the importance of the *choice* of latent representation used in our learning pipeline. Particularly, we show that when the agent's latent representation exhibits a particular form of disentanglement, we can exploit this property to both accelerate the learning of our successor feature matrix and guarantee that GPI finds solutions to certain families of tasks.

Disentangled representations have recently been connected to symmetry transformations (Higgins et al., 2018), a powerful idea borrowed from physics. Roughly speaking, a symmetry transformation for a system is a transformation that leaves some property of the system unchanged. In the context of machine learning, the idea of symmetries is generalised to any structure preserving transformation, like a change in colour, shape, or position of an object. All of these transformations commute with each other in the natural world and can be applied independently, hence forming a symmetry group. This induces an MDP in the state space of discretised disentangled latent dimensions that resembles a hypercube (see Fig. 1), where the nodes on each edge correspond to different values of a single disentangled dimension. Due to the commutative property of disentangled transformations, it suffices to learn how to move along the individual edges of such a hypercube to be able to reach any state inside the hypercube. Hence, by learning a disentangled latent space with $z$ dimensions, and discretising them into $b$ bins, we can re-use $zb$ feature control policies to solve $(b+1)^z$ downstream goal-based tasks, where $b+1$ includes the extra bin value per disentangled dimension that indicates that the value of the corresponding dimension is irrelevant to the task. This intuition is discussed more formally next.

**Optimal independent controllability**  Here we formalise a disentangled representation in terms of features that can be optimally controlled without affecting other features. Let $\phi_{1:n}$ be a set of features and let $\pi_i^*$ be the optimal deterministic policy associated with the control task induced by the cumulant $\mathbf{1}\{\phi_i(s) \in \mathbb{R}_i\}$, with $\mathbb{R}_i \subset \mathbb{R}$. Let $(s_t)_{t=1}^N$ be a sequence of random variables generated by starting in state $s_0$ following $\pi_i^*$. We call $\phi_{1:n}$ *optimally independently controllable* (OIC) if $\mathbb{E}[\phi_j(s_t)] = \phi_j(s_0)$ for all $j \neq i$ and $t \in \{1,...,N\}$.

Note that if we use the features defined in (5) we can have control tasks induced by $\mathbf{1}\{\phi_{ij}(s) = 1\} = \mathbf{1}\{f_i(s) \in b_j\}$ for $j = 1,2,...,m$. In this case two features $\phi_{ij}$ and $\phi_{hl}$ can be OIC only if $i \neq h$. We will abuse the terminology slightly and say that $f_{1:n}$ are OIC if any pair of the induced features $\phi_{ij}, \phi_{hl}$ is OIC when $i \neq h$. Without loss of generality we will assume henceforth that we are using features defined as in (5).

An immediate consequence of a set of OIC features is that values under rewards and policies associated with different features have a simple form:

**Lemma 4.1.** *When $f_{1:n}$ are optimally independently obtainable the successor feature matrix, $\mathbf{\Psi}$, has the following form:*

$$\mathbf{\Psi}_{(lh),(ij)}(s,a) = \begin{cases} \frac{1}{1-\gamma}\phi_{lh}(s,a) & \text{if } i \neq l \\ \Psi_{lh}^{\pi_{ij}^*}(s,a) & \text{else.} \end{cases} \tag{11}$$

*Proof.* The proof follows directly from the definition of OIC features. If $l \neq i$ then under $\pi_{i,j}^*$ $f_l(s_t) = f_l(s_0)$, and thus $\phi_{lh}(s_t,a_t) = \phi_{lh}(s_0,a_t)$, giving:

$$\Psi_{lh}^{\pi_{ij}^*}(s,a) = \mathbb{E}\left[\sum_{t=0}^{\infty}\gamma^t\phi_{lh}(s_t,a_t)\big|\pi_{ij}^*,s_0=s,a_0=a\right] = \mathbb{E}\left[\sum_{t=0}^{\infty}\gamma^t\phi_{lh}(s_0,a_t)\big|\pi_{ij}^*,s_0=s,a_0=a\right]$$
$$= \frac{1}{1-\gamma}\phi_{lh}(s,a). \tag{12}$$

$\square$

Intuitively, Lemma 4.1 states that if the feature $f_i$ associated with policy $\pi_{ij}^*$ is different from the corresponding feature $f_l$ used to define $\phi_{lh}$ the associated successor feature $\mathbf{\Psi}_{(lh),(ij)}(s,a) = \psi_{lh}^{\pi_{ij}^*}(s,a)$ reduces to $(1-\gamma)^{-1}\phi_{lh}(s,a)$. This reduces learning the matrix $\mathbf{\Psi}$ to learning a subset of its entries.

**Guarantees for conjunctions of goal-based tasks**    In addition to simplifying the process of learning the successor feature matrix $\boldsymbol{\Psi}$, features with the OIC property come with guarantees under GPI for certain goal-based tasks. We define a goal-based task as one whose reward function has the form

$$R_G(s) = \mathbf{1}\{s \in G\} \tag{13}$$

where $G \subset \mathcal{S}$. Given the above definition, we say that a policy $\pi$ "achieves" $G$ if $V_{R_G}^\pi(s) > 0$ for all $s \in \mathcal{S}$.

Our uniform discretization of features $f_{1:k}$ into bins $b_{1:m}$ naturally induces a partition over of the state-space:

$$\mathcal{B}(\mathcal{S}) = \{B_{i_1,\dots,i_k} : i_1,\dots i_k \in [m]\} \tag{14}$$

where

$$B_{i_1,\dots,i_k} = \bigcap_{j=1}^{k} f_j^{-1}(b_{i_j}) \subset \mathcal{S}. \tag{15}$$

Intuitively, we can think of each partition $B_{i_1,\dots,i_k}$ as one of the possible $m^k$ configurations of the features $\phi_{ij}$ (note that there are fewer than $2^{mk}$ configurations because some of them are impossible, as two bins associated with the same feature cannot be active at the same time). We can then think of these partitions as goal regions analogous to (13). We now show that for *any* goal $g \in \mathcal{B}(\mathcal{S})$ there exist a linear combination of the cumulants $\phi_{ij}$ that leads to a GPI policy that achieves $g$.

**Theorem 4.1.** *If $f_{1:k}$ are OIC, then for any $g \in \mathcal{B}(\mathcal{S})$ there exists a $w \in \mathbb{R}^{k \times m}$ such that $\pi_w^{GPI}$ as defined in (9) and (10) achieves g. One such $w$ is given by $w_{ij}^g = \mathbf{1}\{f_i(g) = b_j\}$.*

*Proof.* Recall that $\pi_{w^g}^{\mathrm{GPI}} = \mathrm{argmax}_a\ Q_{w^g}^{\max}(s,a)$, where $Q_{w_g}^{\max}(s,a) = \max_{ij} \sum_{lh} w_{lh}^g \Psi_{lh}^{\pi_{ij}^*}(s,a)$. We begin by rearranging terms in $Q_{w^g}^{\max}$:

$$
\begin{aligned}
Q_{w^g}^{\max}(s,a) &= \max_{ij} \sum_{lh} w_{lh}^g \Psi_{lh}^{\pi_{ij}^*}(s,a) \\
&= \max_{ij} \sum_{h=1}^{m} w_{ih}^g \Psi_{ih}^{\pi_{ij}^*}(s,a) + \sum_{h=1}^{m}\sum_{l \neq i} w_{lh}^g \Psi_{lh}^{\pi_{ij}^*}(s,a) \\
&= \max_{ij} \sum_{h=1}^{m} w_{ih}^g \Psi_{ih}^{\pi_{ij}^*}(s,a) + \frac{1}{1-\gamma}\sum_{h=1}^{m}\sum_{l \neq i} w_{lh}^g \phi_{lh}(s,a) \\
&= \max_{ij} \sum_{h=1}^{m} w_{ih}^g \Psi_{ih}^{\pi_{ij}^*}(s,a) + \frac{1}{1-\gamma}\sum_{h=1}^{m}\sum_{l=1}^{k} w_{lh}^g \phi_{lh}(s,a) - \frac{1}{1-\gamma}w_{ih}^g \phi_{ih}(s,a) \\
&= C(s) + \max_{ij} \sum_{h=1}^{m} w_{ih}^g \left[ \Psi_{ih}^{\pi_{ij}^*}(s,a) - \frac{1}{1-\gamma}\phi_{ih}(s,a) \right]
\end{aligned}
\tag{16}
$$

where the third equality follows from Lemma 4.1 and $C(s)$ captures $\phi_{lh}(s,a)$ terms (which do not depend on $a$ or $i$).

First note that, from the form of $w^g$, for each $i$ there is exactly one $j$ such that $w_{ij}^g = 1$ with all other entries being 0. Denote this $j$ as $b(i)$. We can then rewrite:

$$
\begin{aligned}
Q_{w^g}^{\max}(s,a) &= C(s) + \max_{ij}\left[ \Psi_{ib(i)}^{\pi_{ij}^*}(s,a) - \frac{1}{1-\gamma}\phi_{ib(i)}(s,a) \right] \\
&= C(s) + \max_{i}\left[ \Psi_{ib(i)}^{\pi_{ib(i)}^*}(s,a) - \frac{1}{1-\gamma}\phi_{ib(i)}(s,a) \right]
\end{aligned}
\tag{17}
$$

Next notice that $\Psi_{ib(i)}^{\pi_{ib(i)}^*}(s,a) - \frac{1}{1-\gamma}\phi_{ib(i)}(s,a)$ is 0 if $f_i(s) \in b_{b(i)}$ and $\Psi_{ib(i)}^{\pi_{ib(i)}^*}(s,a)$ otherwise, giving:

$$Q_{w^g}^{\max}(s,a) = C(s) + \max_{i \in \mathbb{W}(s)} \Psi_{ib(i)}^{\pi_{ib(i)}^*}(s,a) \tag{18}$$

where $\mathbb{W}(s) = \{i : f_i(s) \notin b_{b(i)}\}$ or, more plainly, the set of feature indices that have not been achieved yet. This gives

$$\pi_{w^g}^{GPI}(s) = \underset{a}{\operatorname{argmax}} \, Q_{w^g}^{\max}(s,a) = \underset{a}{\operatorname{argmax}} \, \underset{i \in \mathbb{W}(s)}{\max} \, \Psi_{ib(i)}^{\pi_{ib(i)}^*}(s,a), \tag{19}$$

implying that $\pi_{w^g}^{\text{GPI}}$ will persistently pursue the "unachieved" feature ($\phi_{lh} = 0$) that is easiest to "achieve" (that is, to be set to $\phi_{lh} = 1$) among the features associated with nonzero elements in $w^g$. This means that eventually all such features will be set to 1, which in turn implies that goal $g$ has been achieved. $\qquad\square$

## 5 SPRITEWORLD EXPERIMENTS

In this section we experimentally validate that an agent can effectively use task-free interactions with an environment to gain a boost in data efficiency across a wide range of subsequent tasks. In particular, we test whether an agent that uses GPI to transfer a set of feature control policies discovered in the ERL setting has a boost in performance over a baseline DQN agent that learns each task from scratch. We also validate whether disentangled feature control policies form a better basis for transfer compared to entangled feature control policies, and whether our pipeline outperforms DIAYN (Eysenbach et al., 2019), a state of the art approach for discovering a diverse set of policies in the absence of external tasks. We chose DIAYN as our baseline, since it is the closest method to ours in terms of motivation (e.g. it does not rely on goal-conditioning in the RL stage unlike Nair et al. (2018)), and is the current published state-of-the-art method in its class (unlike Gregor et al. (2017); Hansen et al. (2019)).

We validate our ideas on a toy Spriteworld environment (Watters et al., 2019) (see Fig. 3). The environment contains an agent and two sprites. The action space is 8-dimensional and consists of moving the agent up/down or left/right, as well as the same four actions but for dragging objects. It is only possible to drag objects if the agent is standing on them. Furthermore, dragging actions move the agent slower than the standard move actions. This environment makes it easy to define a wide range of diverse natural tasks of different difficulty level that can be easily expressed through language. In particular, here we concentrate on a set of navigation based tasks, which specify a goal location for the agent or the objects. The easiest tasks are specified in terms of the final position of the agent ("top", "middle", "bottom" horizontally; and "left", "centre" or "right" vertically). The harder tasks make the locations more specific (by specifying both the horizontal and vertical coordinates, e.g. "top left" or "bottom right"). We also specify equivalent tasks but in terms of the final object position (e.g. "square at the bottom" as an easy object task, or "circle at the top right" as a hard object task). Note that the object-based tasks are more difficult than the agent-based tasks, because the agent position can be controlled

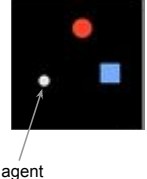

agent

Figure 3: Spriteworld environment. The agent can move up/down, left/right and drag objects when it steps on them.

directly in the action space, while controlling the object position requires more elaborate policies that are also dependent on the agent position. We also specify tasks in terms of disjunctions of agent or object goal locations (e.g. easy tasks like "agent to the left OR square to the top" or hard tasks like "agent to the top left OR square to the top right"). Finally, our hardest tasks are specified as conjunctions of specific locations for both objects (e.g. "square to the top left AND circle to the bottom right"). Note that the goal locations of the downstream tasks may not be directly related to binning specified during the ERL feature control policy learning. Indeed, GPI should be able to generalise the given set of feature control policies to solve a wider range of tasks spanned by the bin boundaries. In the Spriteworld environment the agent receives a reward of 1 if the relevant aspects of the environment state are within their respective goal locations, otherwise the reward is 0. Each episode terminates immediately if the goal is achieved. The agent and the objects are initialised in random positions sampled uniformly within the environment at the start of each episode. We evaluate the performance of our agent and the baselines on 3 tasks sampled from each of the 7 different task classes. Given the structure of our tasks, the average reward corresponds to the average number of episodes on which the agent solves the task.

**Off-Diagonal Trick** As illustrated by Lemma 4.1 when our features $\phi_{1:n}$ are OIC, the matrix of values $\boldsymbol{\Psi}(s,a)$ takes on a specific form where many off-diagonal entries are completely determined by $\phi_{1:n}(s,a)$ and $\gamma$. By assuming that a disentangled representation has the OIC property we can replace these off-diagonal entries by their exact values, both reducing the cost of computing $\boldsymbol{\Psi}$ and associated

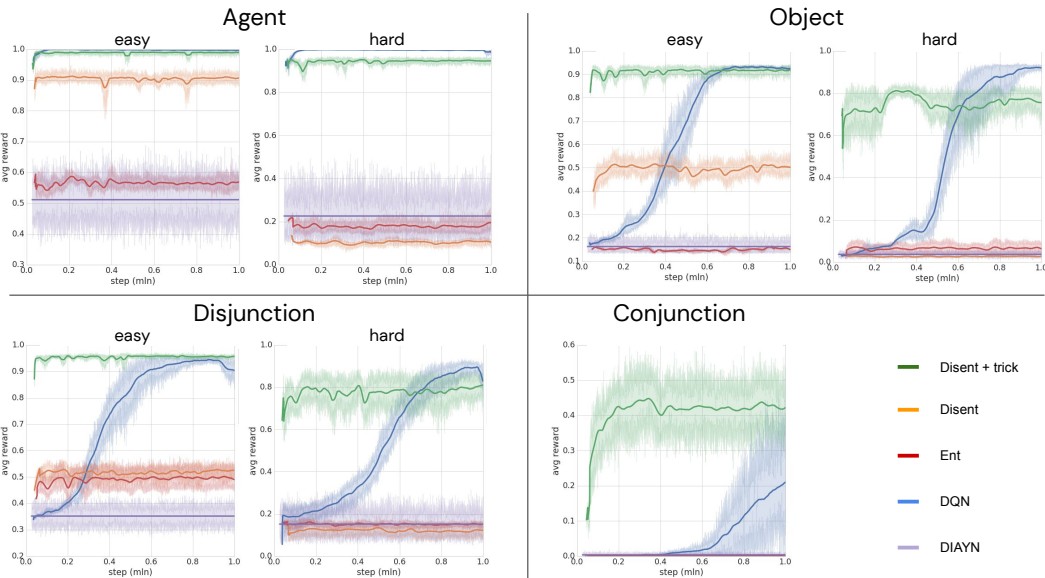

Figure 4: Average reward achieved by the different methods across the same set of tasks. The agent receives a reward of 1 if the goal location is reached, otherwise it receives 0. The rewards are averaged across 3 seeds and 3 tasks per task category. Agent tasks include moving the agent to a specified position. Easy tasks specify a vertical or a horizontal position (e.g. "get the agent to the top"). Hard tasks specify a conjunction of a vertical and a horizontal position (e.g. "get the agent to the bottom right"). Object tasks are similar to agent tasks but specify the goal position of one of the objects (e.g. "get the square to the centre left"). Disjunction tasks are set by specifying a goal in terms of a disjunction of the corresponding easy or hard agent and object tasks (e.g. "get the agent to the left OR get the circle to the middle"). Finally, conjunction tasks are specified as a particular vertical and horizontal position for both objects (e.g. "get the square to the top left AND the circle to the bottom right"). GPI with disentangled features and the off-diagonal trick is able to solve all the tasks at least 50% of the time almost immediately. DQN takes orders of magnitude more steps to achieve similar performance. DIAYN has discovered some agent control policies that allow it to solve some agent based tasks, but it never discovered how to control objects. We used 9 bin feature discretisation to train the feature control policies used by GPI.

error of estimating these off-diagonal terms. We denote the setting of all off-diagonal terms with index $(lh),(ij)$ to $(1-\gamma)^{-1}\phi_{ij}(s,a)$ as the "off-diagonal trick."

**Results** We evaluated how well our approach works in the scenario where disentangled features are the true x and y positions of the agent and the objects, and the entangled features are rotations of the disentangled features. Fig. 4 demonstrates that GPI over feature control policies provides an almost immediate boost in performance over the DQN baseline. This effect gets more prominent as the task difficulty increases, whereby the number of steps before DQN reaches the same performance as the GPI agent increases with task difficulty. The GPI agent is able to solve the tasks most of the time within around 50k learning steps, while the DQN baseline typically requires $> 400k$ steps. We also see that GPI over disentangled features provides a significantly bigger jump in performance compared to GPI over entangled features. Finally, it is clear that the "off-diagonal" trick works well for the disentangled GPI, which suggests an additional benefit of better computational efficiency during ERL learning. Moreover in our comparisons against DIAYN, we found that the competing method could only learn to perform tasks involving the agent, failing to learn to interact with the objects. Finally, Fig. 5 demonstrates that our results are relatively robust over the choice of the disentangled feature bin number ($m = \{5,7,9\}$), and whether the task goal regions are set to the ERL bins exactly (aligned in Fig. 5) or not.

Note that although we have tested our approach on a simple domain using ground truth features, our theoretical results should hold for any complex natural environment and for any natural tasks that can be specified in terms of disentangled transformations and that do not rely on ordered execution. This

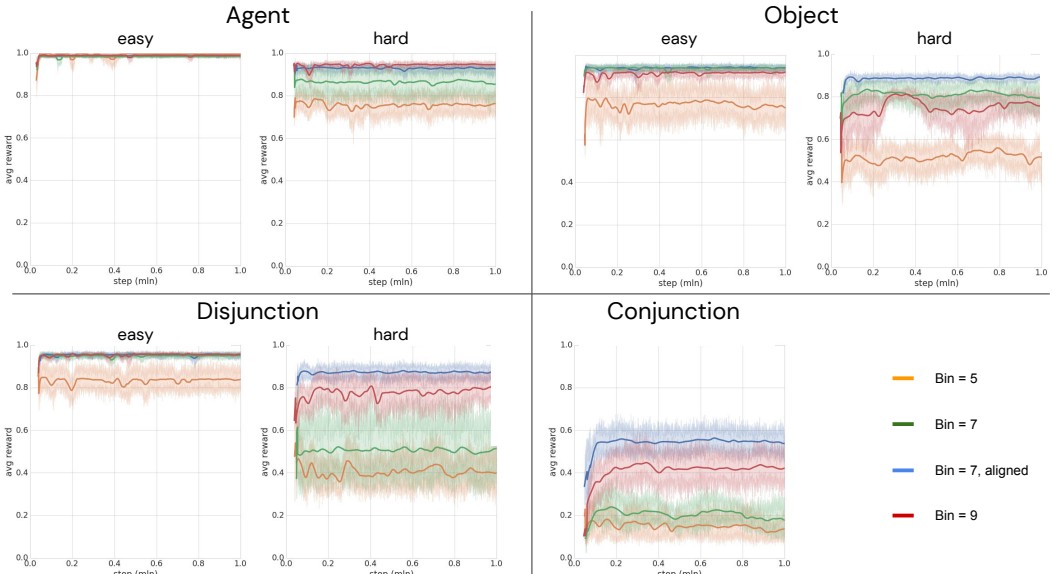

Figure 5: Average reward achieved by GPI with disentangled features and the off-diagonal trick on the same tasks as in Fig. 4, but using different numbers of feature discretisation bins. The performance of our approach decreases with smaller numbers of bins, since GPI has fewer policies to work with, but still significantly outperforms all baselines shown in Fig. 4 in terms of data efficiency. Aligning the downstream tasks with the bin discretisation used to train the feature control policies (bin=7, aligned) slightly improves the performance of our approach, especially on harder tasks.

is because disentangled representation learning aims to find a low-dimensional representation that is equivariant with respect to symmetry transformations, which are a ubiquitous property of our world (Livio, 2012). Hence, any arbitrarily complex natural environment should be possible to describe in terms of a relatively small number of disentangled dimensions. We leave empirical demonstrations on more complex environments using learnt disentangled features to future work.

## 6    CONCLUSIONS

We have proposed a principled way to learn and recombine a basis set of policies that guarantees achievability for a large set of natural tasks that do not require a particular execution order of actions. We have demonstrated that these policies can be learnt in the ERL setting, where the agent has no access to external rewards and has to learn through intrinsically driven interactions with the environment. We theoretically justified and experimentally verified a three-stage pipeline, where the agent 1) discovers disentangled features, 2) learns a set of basis policies through specifying intrinsic rewards when a particular feature achieves a particular range of values, and 3) uses GPI over these policies to bootstrap reasonable performance on a wide range of natural tasks which can be expressed in language and which do not rely on a particular execution ordering. We have empirically demonstrated that GPI over disentangled feature control policies produces better task coverage and learning efficiency on a simple Spriteworld domain compared to a baseline DQN agent and DIAYN, a state-of-the-art method for discovering diverse policies useful for downstream tasks through task-free interactions with the environment. In the future work we would like to generalise our approach to tasks that require a particular order of action execution, as well as to continuous action spaces.

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

# A    APPENDIX

## A.1    SPRITEWORLD ENVIRONMENT

The Spriteworld environment consists of a room without obstacles with an agent and two objects: a circle and a square. The agent can take 8 agents consisting of 4 regular movements (up, down, left, right) and 4 "dragging movements" which mirror the regular movements but move the agent half as far. Objects in the environment can overlap and pass through each other. When the agent overlaps with an object and executes a dragging movement, both the agent and the object are moved together. For our experiments we use a vectorized version of our environment state as observations to our models, consisting of 6 scalars representing the the x and y positions of the agent and each object. When the environment is reset, both the agent and objects are randomly positioned.

## A.2    LEARNING THE SUCCESSOR FEATURE MATRIX

In this section we describe the procedure for learning an approximation of our successor feature matrix $\mathbf{\Psi}$ whose entries are defined by (6). It is helpful to visualize this matrix as follows:

$$\mathbf{\Psi}(s,a) = \begin{bmatrix} \boldsymbol{\psi}^{\pi_{11}^*}(s,a) & \boldsymbol{\psi}^{\pi_{12}^*}(s,a) & \cdots & \boldsymbol{\psi}^{\pi_{mk}^*}(s,a) \end{bmatrix}$$

where each $\boldsymbol{\psi}^{\pi_{ij}^*}$ corresponds to the full vector of successor features evaluated under policy $\pi_{ij}^*$:
$\boldsymbol{\psi}^{\pi_{ij}^*}(s,a) = \begin{bmatrix} \psi_{11}^{\pi_{ij}^*}(s,a) & \psi_{12}^{\pi_{ij}^*}(s,a) & \cdots & \psi_{mk}^{\pi_{ij}^*}(s,a) \end{bmatrix}^\top$. Analogously, we also define the column vector of corresponding cumulants as $\boldsymbol{\phi}(s,a) = \begin{bmatrix} \phi_{11}(s,a) & \phi_{12}(s,a) & \cdots & \phi_{mk}(s,a) \end{bmatrix}^\top$.

We learn a neural network: $Q^\theta(s;g_{ij}): \mathcal{S} \mapsto \mathbb{R}^{|\mathcal{A}| \times m \times k}$. This network maps an environment observations $s$ to a tensor corresponding to $\boldsymbol{\psi}^{\pi_{ij}^*}(s,\cdot)$ — the action-values of the full set of successor features evaluated under $\pi_{ij}^*$. This network is illustrated below:

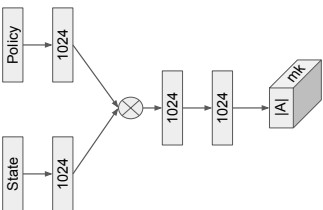

where both environment state $s$ and policy specifier $g_{ij}$ are mapped into $1024$ unit latent representations before being multiplied by each other and then followed by two more $1024$ unit layers before a tensor representing $\boldsymbol{\psi}^{\pi_{ij}^*}(s,\cdot)$ is output. All internal representations are preceded by a leaky-relu activation with hyperparameter $\alpha = 0.1$.

**Behavior**    To deal with the fact that our successor feature matrix encodes multiple distinct policies, given by $\pi_{ij}(s) = \operatorname{argmax}_a Q_{ij}^\theta(s;g_{ij})$, at the beginning of each environment episode we randomly select a feature / bin pair $(i,j)$ and act according to $\pi_{ij}$. When storing experience tuples in our replay buffer, we include the index $ij$ of the policy that generated it.

**Learning**    Largely, we train this network in a manner similar to standard DQN (Mnih et al., 2015) along with dueling and double network architectures (Wang et al., 2016; Van Hasselt et al., 2016) and huber loss for improved stability. We depart from standard architectural choices in how we update each successor feature. For a given experience tuple: $(s,a,s',ij)$ (where $ij$ encodes the policy that generated the transition), we update the entire successor feature vector, $\boldsymbol{\psi}^{\pi_{ij}^*}(s,a)$ by minimizing:

$$\left( \boldsymbol{\psi}^{\pi_{ij}^*}(s,a) - [\boldsymbol{\phi}(s,a) + \gamma \boldsymbol{\psi}^{\pi_{ij}^*}(s',\pi_{ij}^*(s'))] \right)^2.$$

**Sample complexity** On average our agents require around 10 mln steps of interactions with the environment to learn the matrix of successor features. This is then followed by around 50 k steps per downstream task to solve the regression problem specified in Eq. 8 that is required to apply GPI. This is in contrast to $> 500$ k steps per task required by DQN to reach the same level of performance. Hence, the 10 mln steps invested in the ERL stage are a relatively small cost for the data efficiency gains on $(m+1)^k$ downstream tasks, where $m$ is the number of feature bins, and $k$ is the number of disentangled latent dimensions. Indeed, our GPI framework takes the following number of steps to solve all feasible tasks in an environment: $(1000 + 5 * (m+1)^k) * 1e+4$, while the equivalent number for DQN would be $(50 * (m+1)^k) * 1e+4$. Hence, the data efficiency of our approach becomes apparent if the number of downstream tasks is at least 23, since $200 * 5 + 5 * (m+1)^k < 5 * (m+1)^k + 5 * 9 * (m+1)^k$ and $(m+1)^k > 22.2$.

