# OpenReview forum: "Disentangled Cumulants Help Successor Representations Transfer to New Tasks"
_ICLR.cc/2020/Conference — Reject_

### Official Review · AnonReviewer1 · 2019-10-23
**Official Blind Review #1**

**Rating:** 6

**Review:**

The paper addresses the problem of policy transfer in reinforcement learning, which is an extremely relevant open problem in RL, and is being actively studied by the community.  The authors propose a framework for discovering a set of policies without external supervision which can then be used to produce reasonable performance on extrinsic tasks.  The work exhibits originality in that it shows that disentangled representations, learned by intrinsic rewards,  can lead to learn behaviours that are transferable to novel situations.

Although the problem talked here is of high relevance and the approach proposed is original and supported by theoretical results, I am leaning to reject for the following reasons:

- Missing connection to some existing works in the literature. In particular, it seems that there is a link to previous works that focus on discovering reward agnostic options (such as [1]).
- Clarity. The method description is somehow difficult to read, mainly because some variables are introduced without explanation/definition. On page 2, please define n and explain the choices of m and k. The font of the axes and legends in Figure 4 is too small, not readable when printed.
- Experiments: My first concern is that I do not think I would be able to reproduce the results solely given the paper and supplementary material. It would be necessary to either have access to the code, or a very detailed implementation report.
- I would have loved to see either another experiment or at least an intuition on how the framework extends to a very different domain. If not, it should be made clearer that this framework works on 2d domains, where the tasks are navigation tasks. (I am specifically referring to the representation learning phase).

Minor comments:
Page 1, first paragraph “(controlling the position of fruits and nuts)”
page 5, below (9): shouldn’t it be “While \pi_w is not …”?
Page 8, last sentence of Sec 5, “could only learn to perform”.

1. Machado et al, EIGENOPTION DISCOVERY THROUGH THE DEEP SUCCESSOR REPRESENTATION, ICLR 2018



**Experience Assessment:**

I have published one or two papers in this area.

**Review Assessment: Checking Correctness Of Derivations And Theory:**

I assessed the sensibility of the derivations and theory.

**Review Assessment: Checking Correctness Of Experiments:**

I assessed the sensibility of the experiments.

**Review Assessment: Thoroughness In Paper Reading:**

I read the paper at least twice and used my best judgement in assessing the paper.

---

### Official Review · AnonReviewer2 · 2019-10-24
**Official Blind Review #2**

**Rating:** 6

**Review:**

The paper tackles the challenging problem of transfer learning and few shot learning in RL setting and provides some theoretical guarantees for the downstream task coverage.

The paper structure can be further improved by adding a background subsection on successor representation (SR) in RL; SR is not a very well known representation in RL and a brief subsection on that can help the reader in understanding the motivation behind using it. In terms of related work ,another work which can also be mentioned (although not directly related) is “DARLA: Improving Zero-Shot Transfer in Reinforcement Learning” which also uses disentangled representations for zero-shot transfer learning. The paper also needs to be more clear in terms of contributions; it seems that there is a significant overlap between this work and (Barreto et al., 2017, 2018); some clarification would be helpful here.

In terms of empirical results; the authors can also compare with other transfer learning methods in deep RL such as Hansen 2019 or Nair 2018 or explain why these are not reasonable baselines. Also the results for DIAYN are a bit surprising to me since in all the experiments the performance of the method is underwhelming; this is especially surprising because in the original DIAYN paper the method performed well in reasonably complex tasks. Can you provide an intuition on why DIAYN performs poorly even in the agent tasks.

**Experience Assessment:**

I do not know much about this area.

**Review Assessment: Checking Correctness Of Derivations And Theory:**

I did not assess the derivations or theory.

**Review Assessment: Checking Correctness Of Experiments:**

I assessed the sensibility of the experiments.

**Review Assessment: Thoroughness In Paper Reading:**

I made a quick assessment of this paper.

---

### Official Review · AnonReviewer3 · 2019-10-27
**Official Blind Review #3**

**Rating:** 3

**Review:**

This paper proposes to pre-train policies on some goal-reaching tasks, and then leverage the associated successor features to improve the learning of a new task. The method heavily draws from the Generalized Policy Evaluation/Improvement framework without adding much to it. The only relevant point would be showing (as the title indicates) how to obtain disentangled cumulants, and whether they help transfer to new tasks. Nevertheless, both the definition, the full method, and the claimed benefits are quite ambiguous.

Among other concerns showing that the theory needs more formal treatment, the pillar definition of “Optimal independent controllability” is very confusing because it seems to depend on “a trajectory generated by following \pi_i^*”. But what If the environment is stochastic? Then following that policy might give different trajectories! This definition needs to be revisited. More concerning examples are given at the end of this review.

On the experimental side, Fig. 4 is the only reported result, and it has an x-axis that is not clearly explained. What are the “steps (min)”?
It is also not clear what they mean by the “off-diagonal trick”, which seems so important for the good performance.
Furthermore, it seems that their method doesn’t really learn anything new in most of the tasks, it just stays at the same performance that is started with after the whole pre-training steps. It is not clearly stated how much computation effort is required to obtain the desired cumulants, and this invalidates quite strongly any result they report. Even if there’s no “reward” needed during the pre-training, which arguably is not even true because you do need the rewards related to whether you have achieved a specific change in a feature!
In fact, it would be greatly appreciated if the “final” tasks could be expressed in a similar notation than the rest of the pre-training tasks, or vice-versa. As far as I understand, the pre-training tasks consist of making a certain feature fall into a certain subset of its possible values. Can’t the final tasks, like “move the agent to the top right” be also expressed in that form. The link between the two kinds of tasks needs to be much more explicit to be able to assess the relevance of this work.

Finally, they only test their algorithm on Spritworld, which is a small discrete state-action space environment. Even if they try different kinds of tasks in this environment, more detailed analysis or more extensive experiments are needed to assess the benefits of the proposed approach.
This is particularly timely because their method relies on a discretization of some given features that represent the state, which will probably not be very practical in higher dimensional environments.
Finally, I would like a comment on how this method interacts with discrete versus continuous action-state spaces.

Misc comments:
- Why do the authors introduce the terminology “Endogenous RL”, and then say it’s the same as doing RL with intrinsic motivation? This seems like introducing a new name for the same concept, which seems pointless and confusing.
- The connection with “latent learning” of Tolman 1984 is very unclear.
- There’s a “Representation Learning” section, but it’s not clear at all whether any features are actually learned, or whether the features are actually hand-defined. Is the number of features n also hand-defined?
- There might be a typo in the first sentence after equation (9): “While \phi_w is not guaranteed to be optimal with respect to \phi_w”.

Because of all these concerns, I suggest the paper to be weakly rejected.

**Experience Assessment:**

I have published in this field for several years.

**Review Assessment: Checking Correctness Of Derivations And Theory:**

I assessed the sensibility of the derivations and theory.

**Review Assessment: Checking Correctness Of Experiments:**

I carefully checked the experiments.

**Review Assessment: Thoroughness In Paper Reading:**

I read the paper at least twice and used my best judgement in assessing the paper.

---

### Decision · Program_Chairs · 2019-12-19

**Decision:**

Reject

**Comment:**

The author propose a method to first learn policies for intrinsically generated goal-based tasks, and then leverage the learned representations to improve the learning of a new task in a generalized policy iteration framework.  The reviewers had significant issues about clarity of writing that were largely addressed in the rebuttal.  However, there were also concerns about the magnitude of the contribution (especially if it was added anything significant to the existing literature on GPI, successor features, etc), and the simplicity (and small number of) test domains.  These concerns persisted after the rebuttal and discussion.  Thus, I recommend rejection at this time.